# Numerical Simulation of Moisture Diffusion in the Microstructure of Asphalt Mixtures

**DOI:** 10.3390/ma16062504

**Published:** 2023-03-21

**Authors:** Chongzhi Tu, Rong Luo, Tingting Huang

**Affiliations:** 1School of Civil Engineering, Chongqing Jiaotong University, No.66, Xuefu Road, Nan’an District, Chongqing 400074, China; 2Center for High-Performance Materials Research and Development, Gansu Province Transportation Planning, Survery & Designing Institute Co., Ltd., 1685 Yanbei Road, Lanzhou 70030, China; 3Hubei Highway Engineering Research Center, School of Transportation and Logistics Engineering, Wuhan University of Technology, 1178 Heping Avenue, Wuhan 430063, China

**Keywords:** asphalt mxiture, moisture diffusion, water vapour concentrations

## Abstract

Laboratory tests were carried out in accordance with ASTM C664-10(2020) to measure diffusion parameters in an asphalt mixture. These diffusion parameters are effective measures for use in asphalt mixtures. However, the tests could not provide detailed images of the water vapour distribution in the asphalt mixture. To solve this problem, this study provides a method for establishing a numerical model of moisture diffusion in an asphalt mixture, and gives a more detailed picture of the water vapour movement inside the asphalt mixture. Through numerical simulation results obtained under different external conditions, it was demonstrated that water molecules were mainly concentrated in medium III after 100 days of diffusion. After 1000 days of diffusion, the water molecules gradually penetrated the asphalt film in the aggregate particles. Additionally, the water vapour concentrations in the asphalt mixture varied significantly from spot to spot, and various concentration gradients were formed at different locations. These findings revealed that water vapour concentrations in an asphalt mixture vary significantly from spot to spot, forming various concentration gradients at different locations. These findings will be helpful in revealing the mechanism of water damage caused by moisture in asphalt mixtures.

## 1. Introduction

Water vapour diffusion into asphalt mixtures is a major contributor to moisture damage in asphalt pavements, because water vapour may be easily transported into asphalt layers that are impermeable to liquid water [1,2]. Relevant studies have analysed the influence of water vapour on the performance of asphalt mixtures [3,4,5,6,7,8], fine aggregate mixtures [9], and asphalt binders [10,11]. These research results show that water vapour can have a significant impact on the mechanical properties of asphalt mixtures.

To study the movement law of moisture in asphalt mixtures, some research has been carried out based on test designs and established models to measure diffusion parameters, such as diffusivity [12,13,14,15], moisture retention capability [16,17,18,19], and diffusion flux [20,21,22]. The measured diffusion parameters reveal the presence of the diffusion law of water vapour in an asphalt mixture. However, these parameters mainly describe the movement rate of moisture, and they cannot provide a detailed picture of the distribution of moisture in an asphalt mixture. Some researchers use a one-dimensional (1D) diffusion model [19,22] or an empirical diffusion model [23] to describe the total mass of water and gas in asphalt mixtures, but neither is able to characterise the distribution of moisture concentration from spot to spot. As the environment is changing constantly and the internal structure of an asphalt mixture is complex, it is difficult to determine the distribution of water vapour in an asphalt mixture using experiments. Therefore, to study the distribution of water vapour in an asphalt mixture, finite element software is mainly used for simulation. There are two kinds of finite element model for determining water vapour diffusion in an asphalt mixture:(1)A model that regards an asphalt mixture as a homogeneous material [18]. As such, the amount of data is small and the model operation simple. Because an asphalt mixture is a kind of porous material, there are a large number of pores in the mixture, and these pores comprise the main path of water vapour diffusion. Therefore, ignoring the internal structure of an asphalt mixture will cause a simulation result to deviate from the actual situation.(2)A model that regards an asphalt mixture as a combination of a fine aggregate–asphalt mixture (aggregate size ≤ 1.18 mm + pore + asphalt) and coarse aggregate (aggregate size > 1.18 mm) [14,19]. In this type of finite element model, the diffusivities of a fine aggregate–asphalt mixture are equal to the effective diffusivities of an asphalt mixture. As a fine aggregate–asphalt mixture and an asphalt mixture have completely different framework structures, they are unlikely to have the same diffusivity as water vapour [1]. Therefore, using this type of model will lead to a great difference between the numerical results and the experimental results.

To meet these research needs, this study provides a method for establishing a numerical model of the moisture diffusion in an asphalt mixture, and it gives a more detailed picture of the water vapour movement inside the asphalt mixture. The study started with the acquisition of a digital image of an asphalt mixture, which was obtained using a high-definition digital scanner. Then, the digital image of the asphalt mixture was divided into three parts: (1) medium I: coarse aggregate (aggregate particle size > 1.18 mm), (2) medium II: asphalt oil film wrapped on the surface of coarse aggregate, and (3) medium III: a mixture of fine aggregate (particle size ≤ 1.18 mm), pores, and asphalt. In the next section, the process of establishing the finite element model of water vapour diffusion in an asphalt mixture is introduced in detail. The subsequent section presents the moisture distribution in the asphalt mixture at different relative humidities, which were obtained using the finite element model. The final section summarises the major findings of this study and provides a description of ongoing investigations.

## 2. Geometric Model of the Microstructure of the Asphalt Mixture

### 2.1. Preparation of the Asphalt Mixture Specimen

Diffusion experiments were performed on dense-graded asphalt mixture specimens. The specimens were made of limestone aggregates from a quarry located in Fangxian, Hubei Province, China, as well as #70 petroleum asphalt (graded based on the penetration), which was modified using a styrene–butadiene-styrene (SBS) modifier. The gradation of the aggregates is presented in Table 1. The asphalt content was determined to be 4.3%. The asphalt mixture was compacted, using a Superpave gyratory compactor, into raw specimens with dimensions of 150 mm in diameter and 170 mm in height. Each specimen was then cored and cut into a standard specimen with dimensions of 100 mm in diameter and 150 mm in height, and an air void content of 4.3 ± 0.3%. Figure 1 presents examples of a raw specimen and a corresponding standard specimen.

### 2.2. Scanning the Asphalt Mixture Specimen

The main driving force of water vapour diffusion in asphalt is the relative humidity difference between the atmosphere above the asphalt layer and the subgrade under the pavement structure, which leads to water diffusion along the compaction direction of the asphalt mixture [3,4]. Therefore, the geometric image in the finite element model was an external image of the compaction direction of the asphalt mixture.

To obtain a geometric model similar to the actual asphalt mixture structure, an asphalt mixture image system was used to scan an external image of the asphalt mixture. The instrument components and the working principle are shown in Figure 2a, and the equipment setup is shown in Figure 2b. As the asphalt mixture specimen was a cylinder, an ordinary camera could only obtain a plane image, which produced a large error when it was used to establish the model directly. The asphalt mixture image system could continuously image the cylinder’s surface to obtain a complete and high-definition cylinder surface image. The rotation speed of the asphalt mixture was 0.1 r/min, and the maximum resolution reached 2400 dpi. A high-definition picture of the external image was obtained, as shown in Figure 3.

### 2.3. Establishing the Geometric Asphalt Mixture Model

As the image that was acquired by the digital scanner could be transmitted directly, the quality of the image was relatively high. Nevertheless, there were two main problems with the obtained asphalt mixture images:(1)There were many particles in the asphalt mixture, and the particles were dense. Additionally, aggregates with small particles were not clear in the image [24,25].(2)Voids could not be distinguished from asphalt within the images [24].

To ensure good image quality and an accurate finite element calculation amount, in this study, the scanned image was introduced into Photoshop, the contrast and the brightness of the image were adjusted, and the distinction between the aggregate and the asphalt was made as obvious as possible. Then, Image-Pro was used to process the picture, and an outline of the coarse aggregate (>1.18 mm) in the picture was drawn to form the framework of the asphalt mixture, called medium I, as shown in Figure 4. The asphalt film wrapped on the surface of the coarse aggregate was called medium II. The relatively small aggregate (≤1.18 mm) and voids, as well as the asphalt as a whole, were called medium III.

The finite element software used in this study was COMSOL Multiphysics 5.0. The 2D setting in the Model Wizard in the software was selected, the physical field Chemical Specifications Transport > Transport of Divided Specifications (TDS) was added, and Preset Studies > Time Dependent was chosen as the study selection. Additionally, the framework of the asphalt mixture in the Model Toolbar was imported to obtain the numerical figure for the asphalt mixture, as shown in Figure 5.

To establish the numerical simulation of moisture diffusion in the microstructure of the asphalt mixture, the law of moisture diffusion in the asphalt mixture was analysed, and the distribution of the moisture concentration was obtained. The diffusion coefficients of the above three parts needed to be determined.

## 3. Water Vapour Diffusivities of the Asphalt Mixture in the Numerical Model

### 3.1. Water Vapour Diffusivities of Medium I

At present, experimental research on water vapour diffusion is mainly divided into the mass weighing method [26], the thermocouple hygrometer suction measurement method [26,27], the Fourier transform infrared attenuated total reflection method [27], and the electrochemical impedance spectrum method [28]. The data for the mass weighing method are intuitionistic and reliable, and the method does not cause damage to the test piece itself. Compared with other water vapour movement test methods, it can perform direct calculations using the water vapour movement model. Therefore, in this study, the mass weighing method was used to study water vapour diffusion.

Before the water vapour diffusion test was conducted, the limestone parent rock (as shown in Figure 6a) was drilled and cut using a core drill and a cutting saw. For the same parent rock, three cylindrical specimens with diameters of 12 mm and heights of 10 mm were obtained (as shown in Figure 6b). The mass, diameter, and height of the lime test piece were measured, and the test results are shown in Table 2.

A water vapour diffusivity test of the limestone was performed using a gravimetric sorption analyser (IsoSORP^®^ STATIC, RUBOTHERM GMBH, Bochum, Germany), as shown in Figure 7. At the beginning of the test, the limestone specimen was placed in a vacuum environment with a stable temperature (20 °C). The interior of the asphalt mixture test piece was in a dry state after a long period of vacuum pumping. Then, when the water vapour diffusivity test was officially carried out, the limestone test piece was placed in a test environment with a stable temperature and water pressure. Based on the above test conditions, the initial and boundary conditions of the water–gas movement in the asphalt mixture could be determined.

Three limestone specimens were placed in five environments with different relative humidities, and the test conditions are shown in Table 3. The relationship between the mass increment of the test piece and time was determined using the GSA, as shown in Figure 8.

Figure 8 shows that with increasing time, the mass of moisture absorbed by the asphalt mixture test piece increases, but the growth rate gradually slows down. When the moisture movement time is long enough, the level of moisture absorbed by the asphalt mixture test piece will gradually become flat. At this time, the moisture entering the asphalt mixture test piece and leaving the asphalt mixture test piece reaches a dynamic balance, which means that the absorption of moisture by the asphalt mixture at this temperature and water pressure reaches saturation. The higher the external relative humidity, the greater the quality of moisture absorption in the asphalt mixture. However, the time taken for water vapour to reach equilibrium is not much different (approximately 80,000 s).

Fick’s law is used to describe the macroscopic law of the material diffusion phenomenon. Fick’s first law describes steady-state diffusion; that is, *J* and *C* do not change with time. For a one-dimensional condition, the concentration *C* of diffusion components in each diffusion process only changes with distance, not with time. Its expression is as follows:(1)$$J=-D\frac{dC}{dy},$$
where *J* = the diffusion flux (mol/m^2^·s); *D* = the diffusivity (m^2^/s); *C* = the volume concentration of the diffusing substance (mol/m^3^); *dC*/*dy* = the concentration gradient (“–”); the diffusion direction is the reverse direction of the concentration gradient; and *y* = the diffusion distance.

For most diffusion phenomena in nature, diffusion is unsteady; that is, the diffusion phenomenon changes with time. If a time factor is taken into account, then on the basis of the second law of Fick diffusion, the diffusion flux is derived from time *t* to obtain the second law of Fick, which is shown as follows:(2)$$\frac{\partial C}{\partial t}=\frac{\partial }{\partial x}\left(D\frac{\partial C}{\partial x}\right),$$
where *t* = the diffusion time.

Due to small differences in the material distribution, structure, and other aspects of limestone, in this study, it was assumed that the water vapour diffusivities in limestone were the same in all directions. That is, the water vapour diffusivities in limestone were represented by the diffusivities *D*, and the difference between the radial and axial water vapour diffusivities was not distinguished. Through strict mathematical derivation, a model of water vapour diffusion in the cylindrical coordinate system was established, as shown in Equation (3) [5].
(3)$$M\left(t\right)=M\left(\infty \right)\langle 1-\frac{32}{\pi^{2}}\sum_{m=1}^{\infty }\sum_{k=1}^{\infty }\frac{1}{{\left(2k-1\right)}^{2}{\left[x_{m}^{\left(0\right)}\right]}^{2}}\cdot e^{-\left\{\frac{{\left[x_{m}^{\left(0\right)}\right]}^{2}}{a^{2}}+\frac{\left(2k-1\right)\pi^{2}}{H^{2}}\right\}Dt}\rangle ,$$
where M(t) = the quality of the water vapour that diffused into the specimen from the outside and accumulated within it; M(∞) = maximum moisture mass that the test piece could hold; *a* = the section radius of the test piece (mm); *C_0_* = the constant moisture concentration on the surface of the test piece (g/mm^3^); *H* = the height of the test piece (mm); $x_{m}^{\left(0\right)}$ = the root of the Bessel equation of order zero; and the number of terms in the series, with *m_max_* = *k_max_,* was 6, which was the best number of items in the model. In this case, the goodness of fit of the three-dimensional water- and air-motion model for the original test data met the requirements. At the same time, increasing the number of terms did not significantly improve the goodness of fit of the model. In addition, the parameters of each model were stable. The relationship between the moisture concentration and the relative humidity is shown in Equation (4):(4)$$C=RH\cdot \frac{P_{0}m_{H_{2}O}}{RT},$$
where $P_{0}$ = the saturated water vapour pressure at 20 °C and 2338.8 Pa; $m_{H_{2}O}$ = the molar mass of the water vapour (18.015 g/mol); *R* = the universal gas constant (8.314 J/(K·mol)); and *T* = temperature (293.15 K).

By using this model, the original test data (the change in the sample quality with time) that were obtained from the water–gas movement test using the GSA equipment could be processed, and the movement parameters of the water vapour in the asphalt mixture, that is, the diffusivities *D_1_*, could be obtained, as listed in Table 4.

Three observations were made from Table 4:(a).The parameter *M*(*t*) exhibited an excellent relationship with *t*, and every fitting model had an *R*^2^ value larger than 0.90.(b).The fitting curves of the test data of the replicated setups for the same RH differential were fairly close to each other, which demonstrated the repeatability of the designed water vapour diffusion tests.(c).The average water vapour diffusivities of the limestone were 6.7299 × 10^−4^ mm^2^/s. For different relative humidity conditions, the water vapour diffusivities of the same limestone sample showed little difference, and the CV was less than 10%. This shows that the water vapour diffusivities of the limestone change little with relative humidity.

### 3.2. Water Vapour Diffusivities of Medium II

Asphalt is a kind of high-viscosity organic material that is in a liquid state under high temperature or high pressure, so its diffusivities cannot be measured using the GSA. The diffusivity of the asphalt material in this study were measured via FTIR, and were approximately 1.00 × 10^−10^ mm^2^/s [21]. The diffusion of water vapour in the asphalt was not only related to the diffusivities of the asphalt film, but also affected by its thickness. According to the polydispersed sphere systems developed for random heterogenous materials (such as asphalt mixtures) [29,30], the equation for calculating S was a function of the thickness of the asphalt film that covered the aggregate particles, as shown in Equation (5):(5)$$S=\left(1-\phi \right)\left(\alpha_{1}+2\alpha_{2}t+3\alpha_{3}t^{2}\right)\exp\left[-\left(\alpha_{1}+2\alpha_{2}t+3\alpha_{3}t^{2}\right)\right],$$
where ϕ = the volume fraction of the aggregates with diameters greater than the film thickness of the asphalt binder, as shown in Equation (6); t = the film thickness of the asphalt binder (mm); and $\alpha_{1}$, $\alpha_{2}$, and $\alpha_{3}$ = the model parameters, as defined in Equations (6)–(9).
(6)$$\phi =\left(1-VMA\right)\left(1-P_{d}\right),$$
(7)$$\alpha_{1}=\frac{6qm_{2}}{m_{3}},$$
(8)$$\alpha_{2}=\frac{12qm_{1}}{m_{3}}+\frac{18q^{2}m_{2}^{2}}{m_{3}^{2}},$$
(9)$$\alpha_{3}=\frac{8q}{m_{3}}+\frac{24q^{2}m_{1}m_{2}}{m_{3}^{2}}+\frac{16q^{3}m_{2}^{3}}{m_{3}^{3}},$$
where VMA = an acronym for voids in the mineral aggregate, comprised of the volume of intergranular void space between the aggregate particles, which included the air voids and volume of the asphalt not absorbed into the aggregates; $P_{d}$ = the volume fraction of the aggregates passing through the sieve of diameter d (d=t); q = the model parameter as a function of ϕ, ($=\frac{\phi }{1-\phi }$); $m_{1}$ = the mean diameter of aggregate particles; $m_{2}$ = the mean diameter squared; and $m_{3}$ = the mean diameter cubed. Detailed derivation of Equation (5) has been well documented in the literature.

The above equations were applied to asphalt mixtures to calculate the t of each type of asphalt mixture. All of the asphalt mixtures were made of the SBS-modified asphalt binder and the limestone aggregates, which were the same components as those used in the numerical image of the asphalt mixture, as detailed in Section 2. The apparent specific density of the aggregates (without consideration of the permeable voids at the aggregate surfaces) and the bulk specific density of the aggregates (considering both impermeable and permeable voids) were measured, as presented in Table 5. Each batch of aggregates was mixed with the asphalt binder at a temperature of 170 °C; after curing at 150 °C, the loose mix was compacted into a cylindrical specimen with dimensions of 150 mm in diameter and 170 mm in height. The raw specimens were then cored and cut into standard specimens with dimensions of 100 mm in diameter and 150 mm in height. Three replicate specimens were fabricated, and the average properties of the three replicate specimens were tabulated, as shown in Table 6.

### 3.3. Water Vapour Diffusivities of Medium III

#### 3.3.1. Void Characteristics of the Fine Aggregate–Asphalt Mixture

Medium III was formed of a relatively small aggregate (≤1.18 mm) + asphalt + void. Therefore, the aggregate grading of medium III had to be consistent with the fine aggregate–asphalt mixture (FAM), for which the aggregate proportion of the asphalt mixture that passed through a 1.18 mm sieve was equal to the asphalt mixture, as shown in Table 7.

At the same time, the asphalt content of FAM and medium III had to be the same. Using the aggregate-specific surface area method, the asphalt content of medium III was determined. The effective densities *γ* of each grade of aggregate were first measured and calculated, as shown in Table 6. Then, the specific surface area of each grade of aggregate was calculated according to Equation (10), as shown below. If the mass of a grade of aggregate was *M_i_*, the total surface area of the grade of aggregate was calculated according to Equation (11)
(10)$$SSA_{i}=3\left(\frac{1}{\rho_{i}D_{i}}+\frac{1}{\rho_{i-1}D_{i-1}}\right),$$
(11)$$SA_{i}=3M_{i}\left(\frac{1}{\rho_{i}D_{i}}+\frac{1}{\rho_{i-1}D_{i-1}}\right),$$
where $SSA_{i}$ = the specific surface area of the aggregate for sieves $D_{i}$ and $D_{i-1}$; $\rho_{i}$ = the density of the aggregate for sieve $D_{i}$ (g/mm^3^); and $\rho_{i-1}$ = the density of the aggregate for sieve $D_{i-1}$ (g/mm^3^).

Then, the total surface area of the aggregate was obtained by summing the total surface area of the aggregate on each screen. The specific surface area percentage of medium III is shown in Table 8.

It was assumed that the asphalt was evenly distributed on the surface of all of the aggregates, and the asphalt ratio of the fully graded asphalt mixture was 4.3%. According to the same thickness of the asphalt film, we could then obtain the following:(12)$$t=\frac{m_{a}}{\rho_{a}\times SA}=\frac{m_{a}^{ΙΙΙ}}{\rho_{a}\times SA^{ΙΙΙ}},$$
(13)$$P_{b}^{F}=P_{b}\times \frac{m_{a}^{F}}{M^{F}},$$
where $\rho_{a}$ = the asphalt density, g/mm^3^; $SA^{ΙΙΙ}$ = the total specific surface area of medium III; $m_{a}$ = the asphalt quality in the mixture, and $m_{a}^{F}$ = the asphalt quality in medium III. According to Equation (13), the asphalt content of medium III was 17.4%.

In general, the diffusivities of the FAM were used instead of those of medium III. This was the most direct method of measuring the diffusivities of the FAM by forming the specimen. However, the pore structures of the FAM and medium III were completely different, and the pore was the main path of water vapour diffusion, which directly affected the water vapour movement. Therefore, it was necessary to study the relationship between the porosity of the two materials.

According to the gradation and asphalt consumption calculated above, the FAM was formed using a Superpave rotary compactor. Based on the relationship between the compaction height and the porosity, five different porosity specimens were prepared. The mass of each specimen was 3000 ± 100 kg, and the compaction heights were 20 mm, 18 mm, 16 mm, 14 mm, and 12 mm. The compacted test pieces with different porosities are shown in Figure 9a. Then, the drill core was cut into a core sample with a diameter of 12 mm and a height of 20 ± 2 mm, as shown in Figure 9b. The porosity of the core sample was obtained through this test. The measurement results are shown in Table 9.

These results show that the porosity of the FAMs decreased with decreasing compaction height. The moisture diffusion in the asphalt mixture was closely related to its internal void structure. The denser the internal structure of the mixture, the smaller the moisture diffusion flux. According to the volume relationship of each phase of the asphalt mixture, the total volume of the asphalt mixture ($V_{HMA}$) was the sum of medium I, medium II, and medium III ($V_{3}$), as shown in Figure 10. For the asphalt mixture, the thickness of the asphalt film was generally 4–10 μm, so it was assumed that the volume of the asphalt film wrapped on the surface of the coarse aggregate could be ignored and that the porosity of medium III was as follows:(14)$${\varepsilon^{\prime }}_{air}=\frac{V_{air}}{V_{HMA}-V_{c}}$$

After calculation, it was determined that medium III’s porosity was ${\varepsilon^{\prime }}_{air}=16.4\%$, which was approximately five times that of the FAM and four times that of the full graded asphalt mixture. Because the asphalt mixture was a framework structure, there were different sizes of pores in the asphalt mixture under the support of the framework. Medium III was composed of asphalt and fine aggregate, which had a dense structure and was difficult for the FAM. The maximum porosity of the FAM was 3.7%. This meant that medium III, with the same internal structure as the asphalt mixture, could not be formed through the test. However, the moisture diffusion was closely related to the internal void of the asphalt mixture, so in the process of numerical simulation, the diffusivities of medium III could not be directly measured using the test.

#### 3.3.2. Calculation of the Diffusivities of Medium III

(1)Effective diffusivities vs. diffusivities of medium III

The numerical image of the asphalt mixture could be divided into three parts using the image scanning method. The diffusivities of media I and II could be obtained directly through experiments. Due to the special structure and large porosity of medium III, the diffusivities could not be obtained directly through experiments. It was assumed that the diffusion of the asphalt mixture and the three kinds of media in the mixture all satisfied Fick’s second law:(15)$$\frac{\partial C}{\partial t}=\frac{\partial }{\partial y}\left(D_{eff}\frac{\partial C}{\partial y}\right),$$
(16)$$\frac{\partial C}{\partial t}=\sum_{i}^{}\frac{\partial }{\partial y}\left(D_{1}\frac{\partial C}{\partial y}\right)+\sum_{j}^{}\frac{\partial }{\partial y}\left(D_{2}\frac{\partial C}{\partial y}\right)+\sum_{k}^{}\frac{\partial }{\partial y}\left(D_{3}\frac{\partial C}{\partial y}\right),$$
where $D_{eff}$ = the effective diffusivities;$D_{1}$, $D_{2}$, and $D_{3}$ = the diffusivities of media I, II, and III, respectively; and *i*, *j*, and *k* represent the set of points in the three media.

For the same asphalt mixture, the effective diffusivities measured using the the test should have been equal to the results of the numerical simulation. Therefore, the functional relationship between the effective diffusivities of the asphalt mixture and those of medium III can be written as follows:(17)$$\begin{array}{c} \frac{\partial }{\partial y}\left(D_{eff}\frac{\partial C}{\partial y}\right)=\sum_{i}^{}\frac{\partial }{\partial y}\left(D_{1}\frac{\partial C}{\partial y}\right)+\sum_{j}^{}\frac{\partial }{\partial y}\left(D_{2}\frac{\partial C}{\partial y}\right)+\sum_{k}^{}\frac{\partial }{\partial y}\left(D_{3}\frac{\partial C}{\partial y}\right)\\ \;\;\;\;\;\;\;\;\;\;\;\;\;\;\;\;D_{eff}=\frac{D_{1}\sum_{i}^{}\frac{\partial }{\partial y}\left(\frac{\partial C}{\partial y}\right)+D_{2}\sum_{j}^{}\frac{\partial }{\partial y}\left(\frac{\partial C}{\partial y}\right)+D_{3}\sum_{k}^{}\frac{\partial }{\partial y}\left(\frac{\partial C}{\partial y}\right)}{\frac{\partial }{\partial y}\left(\frac{\partial C}{\partial y}\right)}\\ \;\;\;\;\;\;\;\;\;\;\;\;\;\;\;\;D_{eff}=\frac{D_{1}\sum_{i}^{}\frac{\partial }{\partial y}\left(\frac{\partial C}{\partial y}\right)+D_{2}\sum_{j}^{}\frac{\partial }{\partial y}\left(\frac{\partial C}{\partial y}\right)}{\frac{\partial }{\partial y}\left(\frac{\partial C}{\partial y}\right)}+\frac{\sum_{k}^{}\frac{\partial }{\partial y}\left(\frac{\partial C}{\partial y}\right)}{\frac{\partial }{\partial y}\left(\frac{\partial C}{\partial y}\right)}D_{3}\\ \;\;\;\;\;\;\;\;\;\;\;\;\;\;\;\;D_{eff}=a+b\cdot D_{3} \end{array},$$
where *a* and *b* are constants. Therefore, the diffusivities of the asphalt mixture satisfied the linear relationship of medium III. The diffusivities of the asphalt mixture, medium I, and medium II were measured experimentally. Then, function parameters A and B were calculated using finite element software. Finally, the effective diffusivities of the asphalt mixture that were measured by the test were used to determine the actual diffusivities of medium III in the asphalt mixture.

It was assumed that the diffusivities of medium III in the asphalt mixture were in the range of 0.1–1.5 mm^2^/s. Then, they was brought into the finite element software to calculate the corresponding effective diffusivities under five different relative humidity conditions. The calculation results are shown in Table 10. And the relationship between effective diffusivities and diffusivities of medium III is shown in Figure 11.

Through our calculations, the following observations were found:1.The effective diffusivities and medium III diffusivities were fitted with a linear equation. The fitting results are shown in Equation (18). The fitness was 0.9999.
(18)$$D_{eff}=3\times 10^{-8}+6\times 10^{-4}\cdot D_{3}$$

2.The effective diffusivities of the asphalt mixture did not change significantly with the external relative humidity, indicating that the effective diffusivities were mainly affected by the diffusivities of each component and the internal structure.3.The effective diffusivities of the asphalt mixture had an order of magnitude of 10^−4^ mm^2^/s, and the diffusivities of medium III in the asphalt mixture had an order of magnitude of mm^2^/s, with a difference of 10^4^. Therefore, the effective diffusivities were used to directly simulate the moisture diffusion of the asphalt mixture, resulting in serious underestimation of the moisture movement rate.

(2)Diffusivities of medium III in the asphalt mixture

To calculate the diffusivities of medium 3, it was necessary to test the effective diffusivities of the asphalt mixture. The grading curve and materials of the asphalt mixture test piece used in the test were the same as those described in Section 2, and the test method and calculation method were the same as those described in Section 2. Three test pieces were tested, and each was tested under five different relative humidity conditions. The test results are shown in the Table 11.

The average effective diffusivity of the moisture movement of the asphalt mixture was 5.0306 × 10^−4^ mm^2^/s, which was substituted into Equation (18) to obtain the diffusivity of medium III, which was 0.8383 mm^2^/s.

Using the finite element software COMSOL, the diffusivities of medium I, medium II, and medium III were brought into the numerical model. The simulation conditions were set to the test conditions shown in Table 9. After the simulation, the relationship between the water vapour increment of the model and time was derived, as shown in the blue curve in Figure 12. Then, the data obtained in the process of measuring the effective diffusivities of the asphalt mixture were plotted to demonstrate the relationship between the water vapour increment and time, as shown in the red curve in Figure 12. As seen in the figure, the two curves are very similar, with a small deviation. This shows that the finite element model had the same effective diffusion coefficient as the asphalt mixture. It is thus proven that the model had high precision and that it could reflect the water vapour distribution in the asphalt mixture.

## 4. Numerical Model of Moisture Diffusion in the Asphalt Mixture

Water vapour diffusion in an asphalt pavement is an important contributor to moisture damage in the asphalt layer. A major driving force for water vapour diffusion in asphalt is the relative humidity (RH) difference between the atmosphere above the asphalt layer and the subgrade below the pavement structure. The RH in air consistently changes with weather, altitude, and other factors. In contrast, a subgrade serves as a reservoir with RH always above 98%, which is determined using the Kelvin equation based on the fact that the total suction of a subgrade varies from 2 pF to 4.5 pF. The top and bottom boundary conditions for the model were defined in terms of moisture concentration using the RH values shown in Table 12.

No mechanical load was modelled, and the presence of liquid water due to infiltration and capillary rise was ignored. In other words, only vapour diffusion was considered in the simulation. Moisture diffusion was simulated using Fick’s second law. The distribution of the moisture concentration in the asphalt mixture is shown in Figure 13.

Figure 13 shows the evolution of the diffused moisture profile of the asphalt mixture that resulted from the numerical simulation. The closer the colour was to red, the lower the water vapour concentration. The closer the colour was to blue, the higher the water vapour concentration. It can be seen from the figure that with the passage of time, water gradually advances from the top and bottom of the pavement to the centre of the asphalt mixture. This phenomenon is the same as that described in the conclusions of references [15,20].

Moreover, diffusion occurred faster at the bottom of the asphalt mixture because there was higher moisture concentration at that location. The water vapour diffusivities of the asphalt film were much lower than those of medium 3. Therefore, the water molecules were mainly concentrated in medium 3 after 100 days of diffusion of the water in the asphalt mixture. After 1000 days of diffusion, the water molecules gradually penetrated the asphalt film in the aggregate particles. The relationships between the water vapour concentration and time for medium III and medium I (coarse aggregate > 1.18 mm) are shown in Figure 14.

Figure 14 shows that the time required for medium I to reach the maximum water vapour concentration was much longer than that required for medium III. This was because the water vapour always moved preferentially in a medium with large diffusivities, whereas the diffusivities of the asphalt film were very small, which hindered water vapour from penetrating the asphalt film to reach the interior of the aggregate. In addition, the time required for the medium to reach a constant concentration was mainly determined by the diffusivities, which hardly changed with the external relative humidity. To gain a clearer view of the time at which the water vapour concentration inside the medium reached a constant value, the section in the red box in Figure 14 was plotted separately, and is shown in Figure 15.

After 500 h of water vapour diffusion, the water vapour concentration in medium III was essentially in equilibrium. For medium I, after 2000 days of water vapor diffusion, the internal water vapor concentration was in equilibrium. The higher the water vapour concentration, the easier it is to cause the asphalt to peel off from the aggregate surface and cause water damage [13]. Therefore, the above results also show that the internal water damage of the asphalt mixture is more likely to occur in the asphalt mortar. The maximum moisture concentration in the asphalt mixture increased with increasing relative humidity of the upper boundary. This shows that the relative humidity in the air could directly affect the distribution of the moisture concentration in the asphalt mixture.

The numerical simulations demonstrated that the moisture-related material properties of the components of the asphalt mixtures could be efficiently used to track mass transport processes and their related structural degradation processes without the need to conduct complex and expensive experimental work. Therefore, we were able to use these properties as inputs for the numerical model to study the characteristics of the asphalt mixture under different environmental conditions.

## 5. Conclusions

In this study, the diffusivities of water vapour in typical asphalt mixtures were investigated. The mixtures were heterogeneous composite materials made of components with substantially different properties. According to its internal structure, the asphalt mixture was considered to have three media available for water vapour diffusion: (1) a coarse aggregate (>1.18 mm), (2) asphalt film wrapped on the surface of the coarse aggregate, and (3) a small aggregate (≤1.18 mm), and voids of the asphalt as a whole. The diffusivities of media I and II were obtained directly through experiments. Due to the special structure and high porosity of medium III, its porosity was five times that of the FAM. The FAM had the same gradation and asphalt content as medium III, but it had a completely different pore structure. Because the diffusivities of medium III could not be directly measured using an experiment, it was found, via finite element calculation, that the diffusivities of medium III were linear with the effective diffusivities of the asphalt mixture. The order of magnitude of the effective diffusivities of the asphalt mixture was 10^−4^ mm^2^/s, the order of magnitude of the diffusivities of medium I was 10^−4^ mm^2^/s, the order of magnitude of the diffusivities of medium II was 10^−10^ mm^2^/s, and the order of magnitude of the diffusivities of medium III was 10^−1^ mm^2^/s. The diffusivities of medium III were the largest because its porosity was the largest. This reflected the fact that the components of the asphalt mixture had different diffusion properties. Moreover, using the numerical model established in this study, the diffusion data (the relationship between the maximum water mass and the time) were very close to the experimental data, which proved that the model had high accuracy.

The findings of this study suggest that water vapour concentrations in an asphalt mixture vary significantly from spot to spot, which can form various concentration gradients at different locations. In other words, the water molecules were not uniformly distributed in the asphalt mixtures. For the distribution of the moisture concentration in the asphalt mixture for the above model, the water molecules were mainly concentrated in medium III after 100 days of diffusion of water in the asphalt mixture. After 1000 days of diffusion, the water molecules gradually penetrated the asphalt film in the aggregate particles. This finding demonstrates that the asphalt film covering the aggregate surface significantly hindered water vapour diffusion in the aggregates. As the diffusivities of the water vapour in medium III were significantly greater (10^9^ times) than those in the asphalt film, it can be concluded that water vapour was more preferentially diffused in a medium with high diffusivities. The time required for the water vapour to reach a constant concentration in medium III was 500 h, whereas the time required for the water vapour to reach a constant concentration in the aggregate was 2000 days. This further revealed that the complex structure of the asphalt mixture resulted in different parts of the asphalt mixture being affected by water molecules to different degrees.

This study provides a method for establishing a numerical model of moisture diffusion in asphalt mixtures and provides a more detailed image of the movement of water vapour in asphalt mixtures. In addition, it solves the problem in which the water vapour diffusion coefficient of medium III cannot be directly measured by existing tests. Moreover, the model can also track the movement of water vapour in the asphalt mixture. Our numerical simulation can be effectively used to track the water vapour mass transfer process, and then, to analyse the water vapour distribution in an asphalt mixture under different temperatures, humidities, and loads. This numerical model could also be used to analyse water damage in asphalt mixtures in future studies.

## Figures and Tables

**Figure 1 materials-16-02504-f001:**
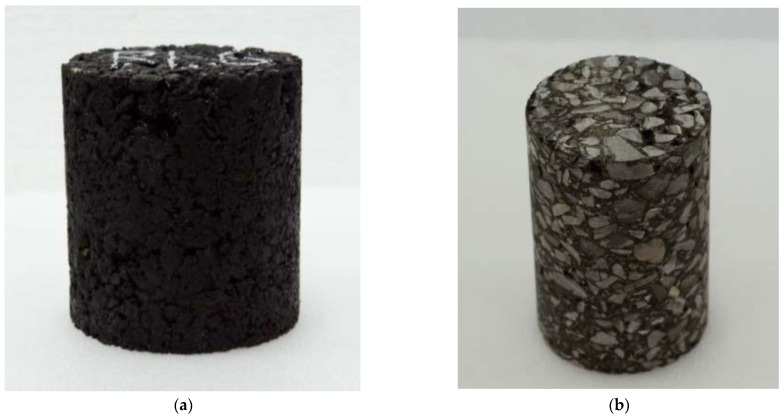
Asphalt mixture test piece: (**a**) raw specimen; (**b**) standard specimen.

**Figure 2 materials-16-02504-f002:**
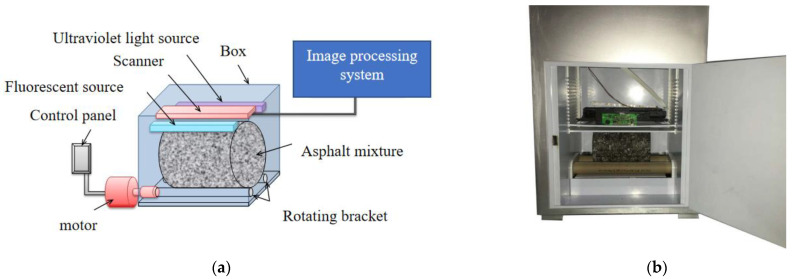
Asphalt mixture image system: (**a**) instrument components and working principle; (**b**) equipment setup.

**Figure 3 materials-16-02504-f003:**
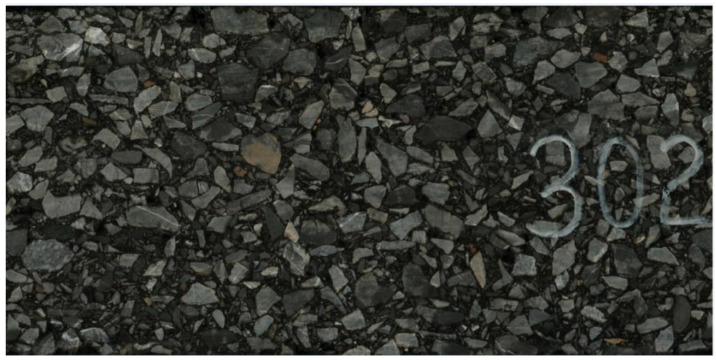
Scanned image of the asphalt mixture.

**Figure 4 materials-16-02504-f004:**
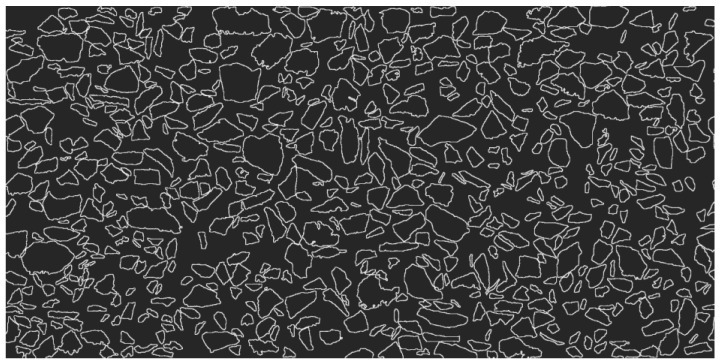
Framework of the asphalt mixture.

**Figure 5 materials-16-02504-f005:**
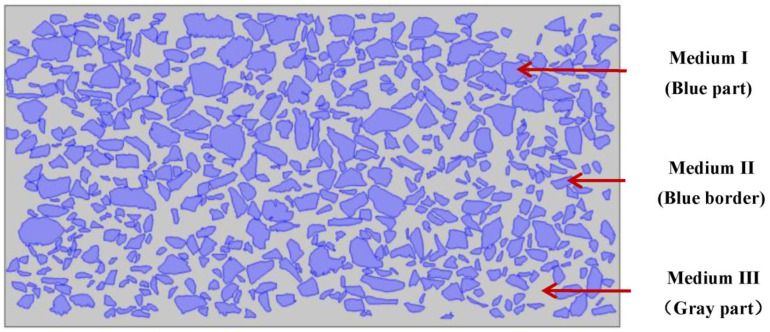
Numerical image of the asphalt mixture.

**Figure 6 materials-16-02504-f006:**
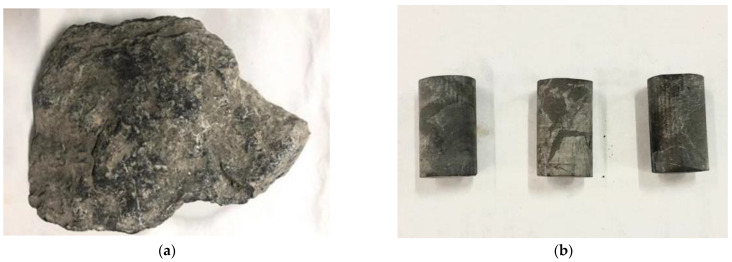
Limestone sample: (**a**) limestone parent rock; (**b**) limestone cylinder test piece.

**Figure 7 materials-16-02504-f007:**
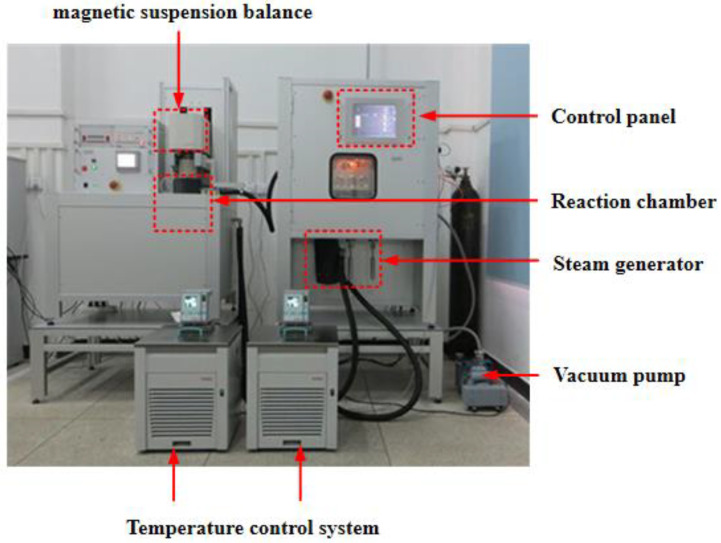
Gravimetric sorption analyser.

**Figure 8 materials-16-02504-f008:**
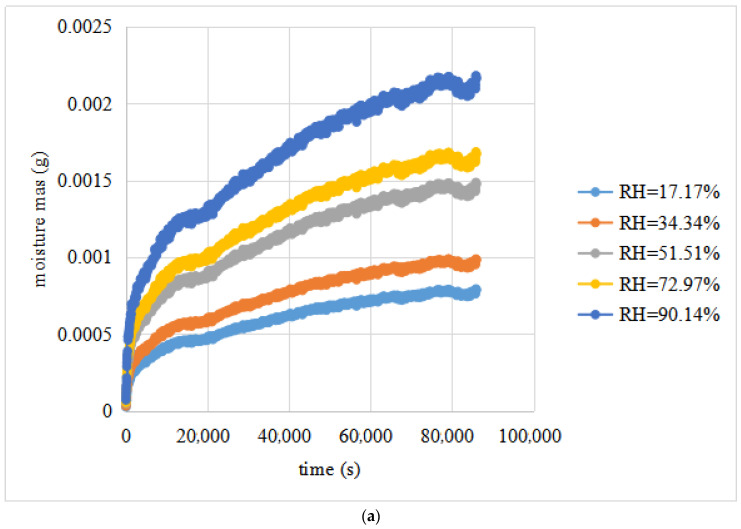
Experimental data for the water–gas diffusion in the limestone: (**a**) original data for the water vapour diffusion of the L-1 test piece; (**b**) original data for the water vapour diffusion of the L-2 test piece; (**c**) original data for the water vapour diffusion of the L-3 test piece.

**Figure 9 materials-16-02504-f009:**
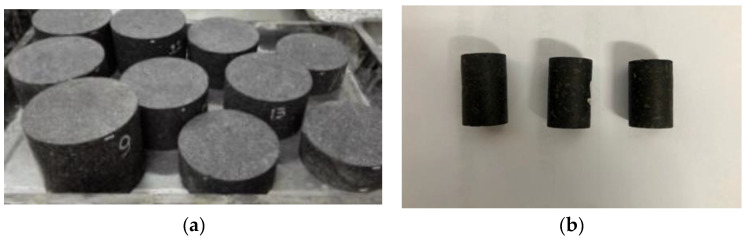
FAM specimens: (**a**) specimens of different heights; (**b**) core sample of medium III.

**Figure 10 materials-16-02504-f010:**
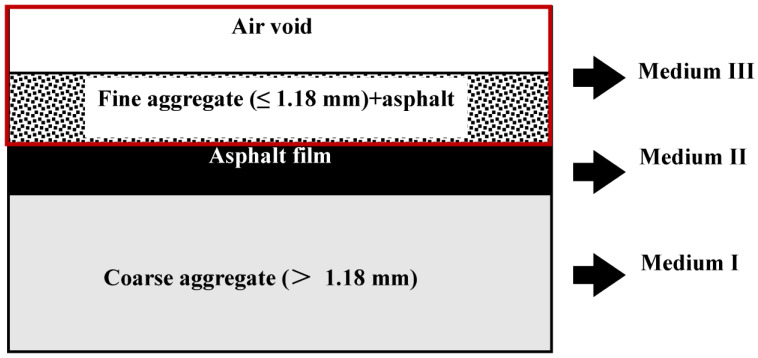
Volume relationship of the asphalt mixture.

**Figure 11 materials-16-02504-f011:**
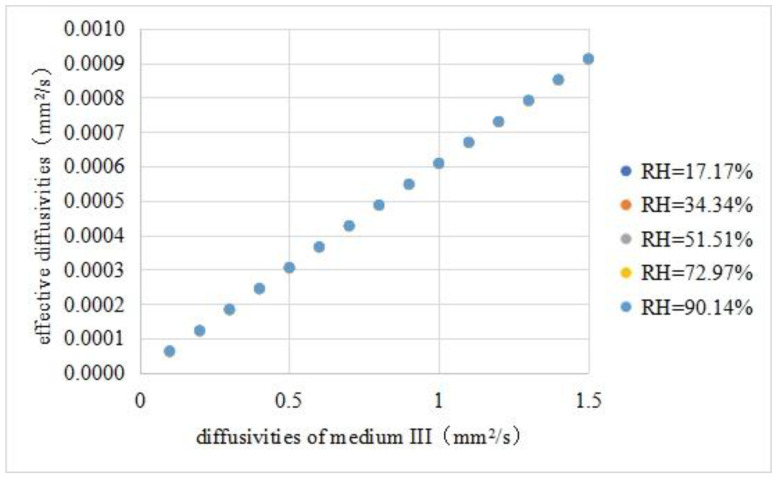
Effective diffusivities vs. diffusivities of medium III.

**Figure 12 materials-16-02504-f012:**
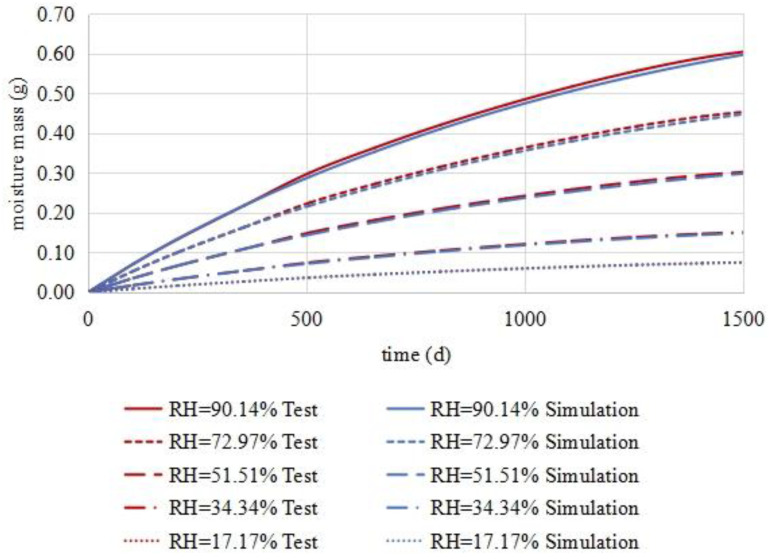
Comparison of the simulation results and the test results.

**Figure 13 materials-16-02504-f013:**
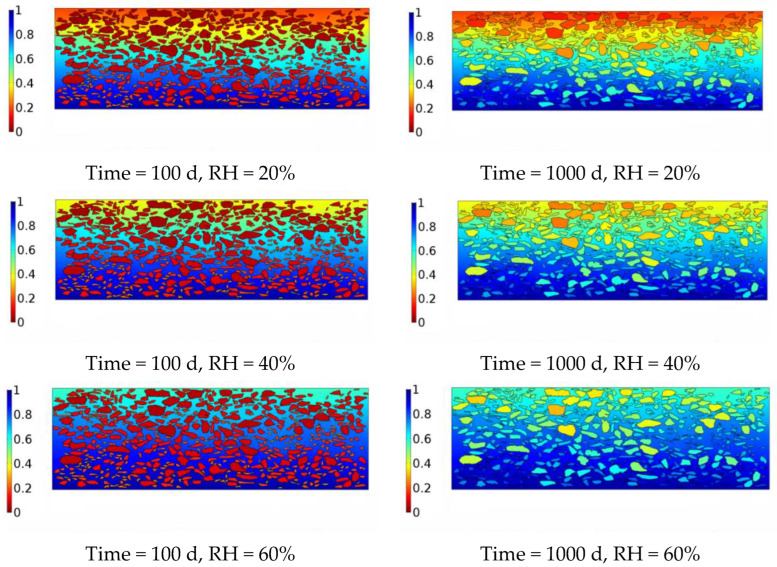
Volume relationships of the asphalt mixture.

**Figure 14 materials-16-02504-f014:**
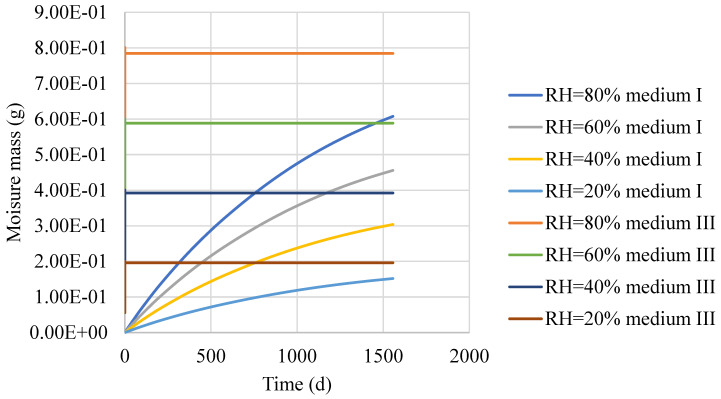
Volume relationships of the asphalt mixture.

**Figure 15 materials-16-02504-f015:**
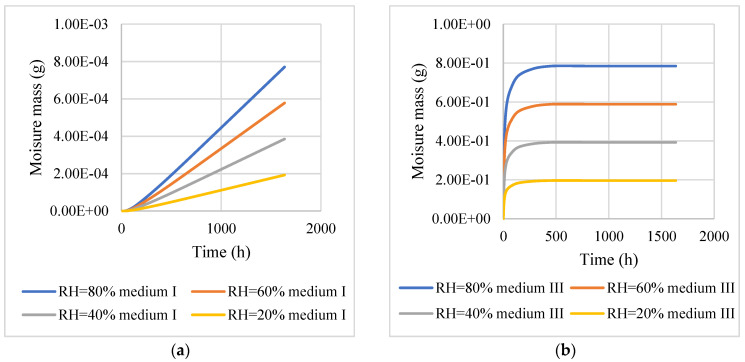
Volume relationships of the asphalt mixture: (**a**) moisture mass in medium I; (**b**) moisture mass in medium III.

**Table 1 materials-16-02504-t001:** Aggregate gradations of five asphalt mixtures.

Sieve Size (mm)	**19**	**16**	**13.2**	**9.5**	**4.75**	**2.36**	**1.18**	**0.6**	**0.3**	**0.15**	**0.075**	**Pan**
Passing (%)	100	97.8	89.5	78.1	54.4	34.6	23.0	18.8	13.7	9.5	7.1	5.5

**Table 2 materials-16-02504-t002:** Diameters, heights, and quality of limestone cylinders.

Set No.	Quality (g)	Diameter (mm)	Height (mm)
L-1	5.2241	12.02	20.01
L-2	5.1276	12.00	19.89
L-3	5.2408	12.04	19.94

**Table 3 materials-16-02504-t003:** Test conditions for the water vapour diffusivities of limestone.

Test Temperature (°C)	Initial Condition		Test Condition	
	Pressure (mbar)	RH (%)	Pressure (mbar)	RH (%)
20	0	0	4	17.17
			8	34.34
			12	51.51
			17	72.97
			21	90.14

**Table 4 materials-16-02504-t004:** Summary of the water and air movement parameters of the asphalt mixture for each relative humidity.

Set No.	RH (%)	*M*(∞) (10^−3^ g)	Diffusivities (*D*_1,_ 10^−4^ mm^2^/s)	Average Diffusivities (10^−4^ mm^2^/s)	CV (%)	*R* ^2^
L-1	17.17	0.2842	7.1038	6.7023	5.51%	0.9328
	34.34	0.8969	6.3121			0.9832
	51.51	1.2921	6.3058			0.9779
	72.97	1.7490	6.9046			0.9729
	90.14	2.5240	6.885			0.9723
L-2	17.17	0.3018	7.4143	6.6614	6.92%	0.9430
	34.34	0.6976	6.7868			0.9804
	51.51	1.1182	6.3749			0.9809
	72.97	1.6589	6.436			0.9802
	90.14	2.2370	6.2952			0.9701
L-3	17.17	0.3240	7.7633	6.8050	8.58%	0.9522
	34.34	0.7566	6.7959			0.9757
	51.51	1.2780	6.4421			0.9845
	72.97	1.8345	6.2448			0.9743
	90.14	2.5196	6.7791			0.9771

**Table 5 materials-16-02504-t005:** Densities of the limestone aggregates used in five types of asphalt mixture.

Sieve Size (mm)	19	16	13.2	9.5	4.75	2.36	1.18	0.6	0.3	0.15	0.075	Pan
Apparent Specific Density	2.734	2.734	2.712	2.712	2.712	2.692	2.758	2.758	2.758	2.758	2.758	2.651
Bulk Specific Density	2.754	2.754	2.749	2.749	2.749	2.769	2.758	2.758	2.758	2.758	2.758	2.651

**Table 6 materials-16-02504-t006:** Properties of asphalt mixtures.

Property	$P_{b}$	$G_{mb}$	VMA	$P_{ba}$	$G_{sb}$	$G_{b}$	$G_{mm}$	$V_{air}$	t	$\bar{r}$
Value	4.3	2.441	14.0	1.0	2.716	1.034	2.596	6.42	4.45	7.486

**Table 7 materials-16-02504-t007:** The grading of the asphalt mixture, FAM, and medium III.

Sieve Size (mm)	Passing of Asphalt Mixture	Passing of FAM	Passing of Medium III
26.5	100		
19	97.8		
16	89.5		
13.2	78.1		
9.5	54.4		
4.75	34.6		
2.36	23	100	100
1.18	18.8	81.74	81.74
0.6	13.7	59.57	59.57
0.3	9.8	42.61	42.61
0.15	7.1	30.87	30.87
0.075	5.5	23.91	23.91
<0.075	0	0	0

**Table 8 materials-16-02504-t008:** Proportion of the specific surface area of the asphalt mixture.

Sieve Size (mm)	Effective Density of Aggregate (g/mm^3^)	*SSA_i_*	*SA_i_*	Percentage
19	2.754	0.10	0.22	6.77%
16	2.754	0.13	1.04	
13.2	2.749	0.15	1.72	
9.5	2.749	0.20	4.68	
4.75	2.749	0.35	6.82	
2.36	2.769	0.70	7.99	
1.18	2.758	1.40	5.80	93.22%
0.6	2.758	2.76	13.95	
0.3	2.758	5.46	21.21	
0.15	2.758	10.88	29.37	
0.075	2.758	21.75	34.81	
<0.075	2.651	39.65	204.25	

**Table 9 materials-16-02504-t009:** Thicknesses and void ratios of the FAM pieces.

Set Compaction Height (mm)	Set No.	Thickness (mm)	Mass (g)	Porosity (%)
20	F-20-1	19.87	5.7910	3.76
	F-20-2	20.02	5.9277	3.82
	F-20-3	19.93	5.8146	3.79
18	F-18-1	19.97	5.8772	3.02
	F-18-2	20.01	5.9267	3.11
	F-18-3	19.81	5.7831	3.07
16	F-16-1	19.81	5.7891	2.34
	F-16-2	20.08	5.8866	2.32
	F-16-3	19.96	5.9591	2.38
14	F-14-1	20.07	5.7049	1.55
	F-14-2	19.95	5.7991	1.49
	F-14-3	20.05	5.7594	1.48
12	F-12-1	19.91	5.8205	1.16
	F-12-2	19.83	5.8747	1.13
	F-12-3	19.92	5.8112	1.14

**Table 10 materials-16-02504-t010:** Functional relationship between the effective diffusivities and the diffusivities of medium III.

Diffusivity of Medium III (mm^2^/s)	Effective Diffusivity of Asphalt Mixture (10^−4^ mm^2^/s)				
	RH = 17.17%		RH = 17.17%		RH = 17.17%
0.1	0.6066	0.1	0.6066	0.1	0.6066
0.2	1.2136	0.2	1.2136	0.2	1.2136
0.3	1.8205	0.3	1.8205	0.3	1.8205
0.4	2.4275	0.4	2.4275	0.4	2.4275
0.5	3.0344	0.5	3.0344	0.5	3.0344
0.6	3.6414	0.6	3.6414	0.6	3.6414
0.7	4.2484	0.7	4.2484	0.7	4.2484
0.8	4.8553	0.8	4.8553	0.8	4.8553
0.9	5.4623	0.9	5.4623	0.9	5.4623
1	6.0692	1	6.0692	1	6.0692
1.1	6.6762	1.1	6.6762	1.1	6.6762
1.2	7.2831	1.2	7.2831	1.2	7.2831
1.3	7.8901	1.3	7.8901	1.3	7.8901
1.4	8.4970	1.4	8.4970	1.4	8.4970
1.5	9.1040	1.5	9.1040	1.5	9.1040

**Table 11 materials-16-02504-t011:** Water vapour diffusion parameters of the asphalt mixture.

Set No.	RH (%)	Effective Diffusivity of Asphalt Mixture (10^−4^ mm^2^/s)	Average Diffusivity (10^−4^ mm^2^/s)	CV	*R^2^*
1	17.17	5.6414	5.0634	7.37%	0.9860
	34.34	4.9004			0.9819
	51.51	4.6283			0.9806
	72.97	5.0128			0.9769
	90.14	5.1341			0.9811
2	17.17	5.2177	5.2843	7.64%	0.9886
	34.34	4.9494			0.9869
	51.51	4.8516			0.9858
	72.97	5.7235			0.9799
	90.14	5.6791			0.9854
3	17.17	4.0449			0.9853
	34.34	4.9004	4.7441	9.13%	0.9833
	51.51	4.6283			0.9911
	72.97	5.0128			0.9801
	90.14	5.1341			0.9941

**Table 12 materials-16-02504-t012:** The calculation parameters for the finite element model.

Property	Value				
Diffusivity	Medium I	6.7229 × 10^−4^ mm^2^/s			
	Medium II	3.60 × 10^−7^ mm^2^/h			
	Medium III	0.8383 mm^2^/s			
Boundary condition	Upper boundary	RH = 20%	RH = 40%	RH = 60%	RH = 80%
	Lower boundary	RH = 100%			
Time step	1 day				
Total calculated duration	1500 days				

## Data Availability

Data openly available in a public repository.

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
