# Peer review of "Numerical Simulation of Moisture Diffusion in the Microstructure of Asphalt Mixtures"

_materials, 2023, doi:10.3390/ma16062504_

Round 1
Reviewer 1 Report
the study abstract should be shorten (250) words. study aim and objectives should be clearly highlited after the introduction. more details about the experimental data must be highlighted after. new references related to study have to be added. also English language must be edited
Reviewer 2 Report
The comments for Authors are compiled in added file.

Reviewer 3 Report
Many thanks for this paper. My comments:
1) INTRODUCTION: please expand your explanation of research gap and potential contributions of this study.
2) LITERATURE REVIEW (BACKGROUND): this section is missing as the authors need to provide a brief description of the main concepts supporting this study
3) METHODOLOGY: I prefer to read a separate methodology section and not one in which results and methods are combined. Please improve.
4) RESULTS: please clarify your results and establish clear differences with the current literature
5) DISCUSSION: analyze results in the light of current literature.
6) CONCLUSIONS:emphasize your contributions and future research paths.
Reviewer 4 Report
The authors presented an article titled: “Numerical Simulation of Moisture Diffusion in the Microstructure of Asphalt Mixtures”. In this paper, the diffusivities of water vapor in typical asphalt mixtures were investigated. The mixtures were heterogeneous composite materials made of components with substantially different properties. This study provided a method for establishing a numerical model of the moisture diffusion of an asphalt mixture, and it gave a more detailed picture of the water vapor movement inside an asphalt mixture. The article is interesting, however, there are several points in the article that require further explanation.
Comment 1:
Abstract
Abstract can be improved. Present in the abstract novelty, practical significance of presented method.
Comment 2:
1. Introduction
There is no cited literature in the text. The introduction should be a discussion of the literature and information about what the authors want to show in the article.
Comment 3:
2. Geometry model of the microstructure asphalt mixture
There is no cited literature in the text. Therefore, it is not known what is new and what has already been published by the authors.
Comment 4:
3. Water vapor diffusivities of the asphalt mixture in the numerical model
There is no cited literature in the text. Provide more information on the test apparatus- Gravimetric Sorption Analyzer (manufacturer, model, etc.).
Correct the font size of the caption under figure 11.
Correct the arrangement of all equations and figures in accordance with the guidelines for authors (the equations and figures are partly on the left margin of the work).
Comment 5:
4. Numerical model of the asphalt mixture moisture diffusion
Correct text editing (figures are partly on the left margin of the work).
Comment 6:
5. Conclusions
Add quantitative and qualitative work results. In addition, it is necessary to more clearly show the novelty of the article and the advantages of the proposed method. What is the difference from previous work in this area? Show practical relevance. Presented conclusions are only a description of the test results. Conclusions should reflect the purpose of the article.
Comment 7:
References
Correct the References in accordance with the guidelines for authors (errors in the notation of literature, for example, there is no volume number of articles).
Comment 8:
The article should be proofread by a native English speaker. The literature is well chosen but is not cited in the text and must be corrected. The article can be valuable and helpful but authors must carefully study the comments and make improvements to the article step by step. Mark all changes in color. After major changes the article can be considered for publication in the “Materials” journal.
Round 2
Reviewer 3 Report
My comments have been resolved.
Reviewer 4 Report
Authors corrected the article. It can be published in Materials in its present form.